# The Role of Primary Tumor Resection in Colorectal Cancer Patients with Asymptomatic, Synchronous, Unresectable Metastasis: A Multicenter Randomized Controlled Trial

**DOI:** 10.3390/cancers12082306

**Published:** 2020-08-16

**Authors:** Eun Jung Park, Jeong-Heum Baek, Gyu-Seog Choi, Won Cheol Park, Chang Sik Yu, Sung-Bum Kang, Byung Soh Min, Jae Hwang Kim, Hyeong Rok Kim, Bong Hwa Lee, Jae Hwan Oh, Seung-Yong Jeong, Minkyu Jung, Joong Bae Ahn, Seung Hyuk Baik

**Affiliations:** 1Division of Colon and Rectal Surgery, Department of Surgery, Gangnam Severance Hospital, Yonsei University College of Medicine, Seoul 06273, Korea; camp79@yuhs.ac; 2Department of Surgery, Gil Medical Center, Gachon University College of Medicine, Incheon 21565, Korea; gsbaek@gilhospital.com; 3Colorectal Cancer Center, Kyungpook National University Chilgok Hospital, School of Medicine, Kyungpook National University, Daegu 41404, Korea; kyuschoi@mail.knu.ac.kr; 4Department of Surgery, Wonkwang University School of Medicine, Iksan 54538, Korea; parkwc@wku.ac.kr; 5Department of Colon and Rectal Surgery, Asan Medical Center, University of Ulsan College of Medicine, Seoul 05505, Korea; csyu@amc.seoul.kr; 6Department of Surgery, Seoul National University Bundang Hospital, Seoul National University College of Medicine, Seongnam 13620, Korea; kangsb@snubh.org; 7Division of Colon and Rectal Surgery, Department of Surgery, Severance Hospital, Yonsei University College of Medicine, Seoul 03722, Korea; BSMIN@yuhs.ac; 8Department of Surgery, Yeungnam University College of Medicine, Daegu 42415, Korea; jhkimgs@ynu.ac.kr; 9Department of Surgery, Chonnam National University Hwasun Hospital and Medical School, Hwasun 58128, Korea; drkhr@jnu.ac.kr; 10Department of Surgery, Hallym University Sacred Heart Hospital, Anyang 14068, Korea; a15211@daum.net; 11Center for Colorectal Cancer, Research Institute and Hospital, National Cancer Center, Goyang 10408, Korea; jayoh@ncc.re.kr; 12Department of Surgery, Seoul National University College of Medicine, Colorectal Cancer Center, Seoul National University Cancer Hospital, Seoul 03080, Korea; syjeong@snu.ac.kr; 13Division of Medical Oncology, Department of Internal Medicine, Yonsei Cancer Center, Yonsei University College of Medicine, Seoul 03722, Korea; MINKJUNG@yuhs.ac (M.J.); VVSWM513@yuhs.ac (J.B.A.)

**Keywords:** primary tumor resection, colorectal neoplasm, synchronous unresectable metastasis, non-curative resection, neoplasm metastasis, chemotherapy, overall survival

## Abstract

We aimed to assess the survival benefits of primary tumor resection (PTR) followed by chemotherapy in patients with asymptomatic stage IV colorectal cancer with asymptomatic, synchronous, unresectable metastases compared to those of upfront chemotherapy alone. This was an open-label, prospective, randomized controlled trial (ClnicalTrials.gov Identifier: NCT01978249). From May 2013 to April 2016, 48 patients (PTR, *n* = 26; upfront chemotherapy, *n* = 22) diagnosed with asymptomatic colorectal cancer with unresectable metastases in 12 tertiary hospitals were randomized (1:1). The primary endpoint was two-year overall survival. The secondary endpoints were primary tumor-related complications, PTR-related complications, and rate of conversion to resectable status. The two-year cancer-specific survival was significantly higher in the PTR group than in the upfront chemotherapy group (72.3% vs. 47.1%; *p* = 0.049). However, the two-year overall survival rate was not significantly different between the PTR and upfront chemotherapy groups (69.5% vs. 44.8%, *p* = 0.058). The primary tumor-related complication rate was 22.7%. The PTR-related complication rate was 19.2%, with a major complication rate of 3.8%. The rates of conversion to resectable status were 15.3% and 18.2% in the PTR and upfront chemotherapy groups. While PTR followed by chemotherapy resulted in better two-year cancer-specific survival than upfront chemotherapy, the improvement in the two-year overall survival was not significant.

## 1. Introduction

Colorectal cancer (CRC) is the third most common cancer and the second leading cause of cancer-related death according to the global cancer statistics [1]. Patients with stage IV CRC who are diagnosed with distant metastasis at initial CRC diagnosis make up 20% of the total cases of CRC [2]. In the treatment of stage IV CRC, primary tumor resection (PTR) has been performed to relieve tumor-related complications and to avoid life-threatening conditions such as intractable bleeding, intestinal obstruction, and perforation. However, it is still controversial whether upfront PTR is necessary in patients with asymptomatic, synchronous, unresectable metastatic CRC.

PTR in patients with asymptomatic stage IV CRC can prevent impending obstruction and emergent situations caused by the primary cancer. Since these complications are associated with increased mortality and morbidity, upfront PTR is preferred to avoid tumor-related complications, which can develop during chemotherapy. In addition, a decrease in tumor burden after PTR is expected to increase the potential survival. Several retrospective studies and meta-analyses have reported survival benefits of PTR compared with upfront chemotherapy [3,4,5]. According to the analysis of the Surveillance, Epidemiology, and End Results (SEER) database by Tarantino et al. [6,7], PTR results in prolonged overall and cancer-specific survival compared to upfront chemotherapy. In addition, a nationwide study in the Netherlands found that PTR followed by systemic chemotherapy showed survival benefits compared with chemotherapy alone [8].

In contrast, upfront chemotherapy is also widely used for asymptomatic stage IV CRC as an initial treatment. The advance of modern systemic chemotherapy using combined chemotherapy with molecular target agents increased the survival rate of patients with metastatic CRC dramatically, from six months to nearly 24 months [9,10,11,12,13]. The time-trend analysis of patients with stage IV CRC in the United States demonstrates that survival has increased in spite of a decreased rate of PTR in the modern era [14]. In addition, delayed initial chemotherapy and potential risks of postoperative complications are regarded as drawbacks of PTR as the initial treatment in patients with asymptomatic metastatic CRC.

The survival benefits of PTR in patients with asymptomatic metastatic CRC remain controversial. The oncologic effects of PTR are difficult to validate due to a lack of results from randomized controlled trials and limitations of retrospective data from a heterogeneous patient selection. Therefore, this study aimed to evaluate whether PTR followed by chemotherapy, improves survival and decreases complication rates compared with upfront chemotherapy in patients with asymptomatic, synchronous, unresectable metastatic CRC in a multicenter randomized controlled trial.

## 2. Results

The study period was conducted from May 2013 to April 2016. It ended early due to a lack of patient enrollment and cessation of funding. Patients were recruited from October 2013 to May 2015. The final data collection was on 30 April 2016. The median follow-up period was 15.0 months (interquartile range (IQR), 8.5–21.5 months). A total of 52 patients from 12 hospitals in Korea were randomized and allocated into two groups: the upfront chemotherapy only group (Arm 1, *n* = 25) and the PTR with chemotherapy group (Arm 2, *n* = 27), as shown in Figure 1. According to the intention-to-treat protocol, 22 patients from Arm 1 and 26 patients from Arm 2 were evaluated. Four patients who did not receive treatment were excluded. Four more patients were lost to follow-up, resulting in a final count of 21 patients in Arm 1 and 23 patients in Arm 2.

### 2.1. Baseline Patient Characteristics

There were no statistical differences in age, sex, body mass index, American Society of Anesthesiologists (ASA) score, performance status, or co-morbidities between the PTR group and the upfront chemotherapy group. The sigmoid colon was the most frequent primary cancer site in both groups. Liver metastasis occurred in 50.0% of patients in the upfront chemotherapy group and 61.5% of patients in the PTR group. Synchronous multiple metastases were present in 45.5% of the upfront chemotherapy group and in 23.1% of the PTR group, but this difference was not statistically significant (*p* = 0.320). A single metastasis was present in 54.5% of the upfront chemotherapy group and in 76.9% of the PTR group. The initial carcinoembryonic antigen level was not significantly different between the groups (Table 1).

### 2.2. Clinical Manifestations of Upfront Chemotherapy Group (Arm 1)

The conversion rate of initially unresectable metastatic tumors to resectable tumors was 18.2% in the upfront chemotherapy group, as shown in Table 2. Liver metastases were reduced in size, rendering them resectable in four patients (18.2%) during palliative first-line chemotherapy using FOLFIRI with target agents (two patients with cetuximab and two patients with bevacizumab). Four patients (18.2%) underwent operations during chemotherapy. Two patients underwent palliative PTR, and two patients underwent surgery to treat intestinal obstruction. In the upfront chemotherapy group, five patients (22.7%) required interventional therapies—three gastrointestinal stent insertions (13.6%), and two salvage radiotherapies (9.1%).

### 2.3. Clinical Manifestations of the PTR Group (Arm 2)

PTR was performed to resect the primary tumor completely before chemotherapy. Anterior resections and low anterior resections were each performed in eight (30.8%) patients. Sixteen open surgeries (61.5%) were performed, compared to 10 laparoscopic surgeries (38.5%). Three patients (11.5%) underwent synchronous operations during PTR. The mean length of hospital stay after the PTR operation was 11.8 ± 6.7 days (range, 6–34 days). Three patients were re-admitted within 30 days post-operation due to septic shock, general weakness, or postoperative ileus. One patient (3.8%) died within 30 days postoperatively due to pneumonia with sepsis. The initial adjuvant chemotherapy after PTR was started within 24.8 ± 11.4 days of PTR. The pathology reports showed that 65.4% of patients had T3 tumors, and that 88.5% of the primary tumors were moderately differentiated. The mean tumor size was 6.0 cm, and the mean number of positive regional lymph nodes was 5.0. Nineteen patients (73.1%) had a lymphovascular invasion. The conversion rate of initially unresectable metastatic tumors to resectable tumors was 15.3% in the PTR group (Table 3).

### 2.4. Regimens for Chemotherapy

The most common first-line chemotherapy used in both groups was the FOLFIRI regimen (73.1% of the PTR group patients and 90.9% of the upfront chemotherapy group patients). Anti-Vascular endothelial growth factor(VEGF) therapeutics were used more frequently in the upfront chemotherapy group than in the PTR group. FOLFOX chemotherapy was the most used second-line regimen for palliative chemotherapy (53.8% of the PTR group patients and 45.5% of the upfront chemotherapy group patients). Various regimens were used as third-line chemotherapies (Appendix A).

### 2.5. Complications Between PTR and Upfront Chemotherapy Groups

In the upfront chemotherapy group, the primary tumor-related complication rate was 22.7%. Three patients (13.6%) experienced grade IIIa intestinal obstructions, and two patients (9.1%) had grade IIIb intestinal obstructions. Six patients (27.3%) experienced grade I chemotherapy-related toxicities. Postoperative complications developed in five patients (19.2%) after PTR. Only one major postoperative complication (3.8%) occurred, as one patient had a grade V complication due to pneumonia with sepsis after PTR. The rate of PTR-related grade I–II complications was 15.4%. There was no anastomotic leakage after PTR in this study. Chemotherapy-related toxicities accounted for 26.9% of grade I complications in the PTR group. One patient had a grade IV complication caused by pneumonia during chemotherapy (Table 4).

### 2.6. Survival Outcomes

The two-year overall survival rate of the PTR group was not significantly higher than that of the upfront chemotherapy group, despite a drastic difference (69.5% in the PTR group and 44.8% in the upfront chemotherapy group, *p* = 0.058). However, the two-year cancer-specific survival rate of the PTR group was significantly higher than that of the upfront chemotherapy group (72.3% in the PTR group and 47.1% in the upfront chemotherapy group, *p* = 0.049) (Figure 2).

## 3. Discussion

In this trial, PTR followed by systemic chemotherapy showed improved two-year cancer-specific survival in patients with asymptomatic CRC with synchronous unresectable metastases compared with upfront chemotherapy alone. However, the two-year overall survival rate after PTR plus chemotherapy was not statistically significant compared with upfront chemotherapy. Although major postoperative complications occurred in 3.8% of patients after PTR, 22.7% of patients experienced major tumor-related complications during upfront chemotherapy.

The goal of PTR in patients with asymptomatic stage IV CRC is to prevent primary tumor-related complications. Although the tumors are asymptomatic at diagnosis, life-threatening tumor-related complications, including massive tumor bleeding, bowel obstruction, or perforation, can occur as the disease progresses. In addition, chemotherapy-related myelosuppression decreases the potential recovery, increasing the mortality in these emergent situations. In contrast, there are reports that the rates of emergent interventions are relatively low in patients using combination chemotherapy (11–14%) [15,16]. In a retrospective study by Poultsides et al. [15] involving patients with synchronous stage IV CRC receiving triple-drug chemotherapy with or without bevacizumab as the initial treatment, 7% of patients required emergent operations due to primary cancer obstruction or perforation and 4% of patients required therapeutic interventions. McCahill et al. [16] reported that the 24-month cumulative rate of major morbidities in patients undergoing chemotherapy with mFOLFOX6 plus bevacizumab without PTR was 16.3%. In our study, 22.7% of patients developed primary tumor obstruction during upfront chemotherapy. Three of these patients (13.6%) were treated with colonic stent insertion, and two patients (9.1%) underwent emergent surgery. The primary tumor-related complications observed in this study are comparable to those previously reported in retrospective studies and are all major morbidities at or above grade III complications. In contrast, major complications occurred in 3.8% of patients after PTR in this study.

In the clinical setting, it is difficult to predict the necessity of therapeutic interventions during chemotherapy in patients with asymptomatic metastatic CRC. The asymptomatic tumor of each patient differs in size, depth of invasion, and rate of circumferential narrowing of the bowel lumen. These differences make the decision to perform PTR prior to chemotherapy challenging. Regular endoscopic surveillance and categorization of asymptomatic tumors into subgroups based on the risks of primary tumor-related complications are useful tools to determine whether PTR should be performed in patients with asymptomatic stage IV CRC.

In this trial, the two-year cancer-specific survival rate after PTR followed by chemotherapy was higher than that of chemotherapy alone. There were concerns that delaying systemic chemotherapy for PTR would negatively affect survival due to the slow recovery and potential postoperative complications. However, the time interval to begin chemotherapy after PTR was 24.8 postoperative days, and the mean hospital stay was 11.8 days, which is comparable with major colorectal resection procedures [17,18]. Although one patient had difficulties beginning chemotherapy due to major complications of pneumonia with sepsis after PTR, most patients received chemotherapy without difficulty. Most of the PTR-related complications were limited to grade I–II complications. Some retrospective analyses have suggested that patients who undergo PTR have better initial performance status, which may confound the results of the two-year survival rates [19]. However, our trial was randomized, and the results showed that the two-year cancer-specific survival of the PTR group was better than that of the upfront chemotherapy group. Fast recovery with fewer postoperative complications and the prevention of potential tumor-related complications during chemotherapy are some advantages of PTR that may play a role in improving cancer-specific survival.

In contrast, it is unclear whether PTR has benefits for overall survival. Many retrospective and meta-analyses reported the survival benefits of PTR, followed by chemotherapy compared with chemotherapy alone [4,20,21]. However, in our trial, the PTR group did not have a statistically significant survival benefit compared to the upfront chemotherapy group, despite a much higher survival rate. The difference observed in our study is comparable with the result of the first interim analysis of the JCOG1007 study, which is a randomized phase III trial comparing PTR plus chemotherapy and chemotherapy alone [22,23]. The median overall survival of the trial was 25.9 months in patients who underwent PTR plus chemotherapy and 26.7 months in patients who underwent chemotherapy alone (*p* = 0.69) [23]. Patient survival rates have improved with the development of new chemotherapeutic agents. The 17.4-month overall survival observed in patients receiving FOLFIRI monotherapy increased to 25.0 months in patients receiving combined chemotherapies with bevacizumab and to 28.7 months in patients receiving cetuximab, as reported in the FIRE-3 trial [24,25]. The National Cohort Study of the US population from 1988 to 2010 showed that the overall survival of stage IV CRC improved despite a decreasing rate of PTR [14]. Thus, it was believed that fewer surgeries are required to improve survival. However, after a propensity score matching analysis of this database to remove confounding factors, the rates of the overall and cancer-specific survival increased simultaneously in patients who underwent PTR and in patients who received upfront chemotherapy [6]. The development of novel chemotherapeutic agents increased the overall survival in patients who received PTR followed by chemotherapy as well as in patients who received chemotherapy alone. The rates of conversion to resectable metastases were comparable between the groups of this study—18.2% in the upfront chemotherapy group and 15.3% in the PTR group. Despite the hypothesis that surgical stress and inflammatory conditions after PTR may negatively impact survival, a good response to chemotherapeutics was observed in both groups regardless of PTR. Although current clinical guidelines do not recommend palliative PTR, the overall survival benefit of PTR followed by chemotherapy should be investigated comprehensively with consideration for treatment response and potential risks from primary tumor progression.

Although this study tried to evaluate the role of PTR in the treatment of unresectable metastatic CRC, there are several inevitable limitations. First, the early termination of the study resulted in a lack of data and short follow-up periods, which led to low statistical power. Difficulties in the study enrolment process and the abrupt cessation of research funding were obstacles to the completion of this study. In particular, performing the operations and reaching the final goal successfully are more difficult in clinical trials involving a surgical intervention than in those involving a medical intervention. Active participation regarding undergoing surgical interventions such as PTR was challenging for the patients as well as the surgeons. In addition, continued research funding and support while awaiting the final results are required to conclude the study. Second, we did not assess the quality of life, which is an important factor for selecting treatment methods for stage IV CRC. Third, the rate of left-sided colon cancer was higher than right-sided colon cancer in both groups. In addition, patients who had peritoneal metastases, which leads to bowel obstruction more frequently than metastasis to other sites, were excluded. Fourth, although most patients in this study were treated with the recommended chemotherapeutic regimens, some patients were unable to receive this treatment. Lastly, we did not perform genetic analyses to identify patients with incurable stage IV CRC who might benefit from PTR. Nevertheless, the results of this study are crucial, as we evaluated the oncologic impact of PTR in a homogeneous study population in which confounding factors were decreased due to randomization. A low rate of postoperative complications and the potential to prevent possible complications of primary tumors without delaying chemotherapy make PTR a beneficial treatment option in patients with stage IV CRC. Randomized controlled trials for PTR in patients with incurable stage IV CRC—SYNCHRONOUS (ISRCTN trial registry: 30964555) and CAIRO4 (ClinicalTrials.gov Identifier: NCT01606098)—are ongoing [22,26,27]. It is expected that the role of PTR will be well defined when the comprehensive results of these trials are reported in the near future.

## 4. Materials and Methods

### 4.1. Study Design

This was an open-label, prospective, randomized controlled trial that took place in 12 tertiary hospitals in Korea. All patients were screened, then randomized in a 1:1 ratio into one of the two arms—Arm 1 (upfront chemotherapy alone) and Arm 2 (chemotherapy after PTR). Patients allocated into Arm 1 received chemotherapy without PTR within two weeks after randomization. Patients of Arm 2 received PTR within two weeks after randomization and then began to receive chemotherapy within two months after PTR. PTR was defined as the complete resection of the primary tumor (R0 resection) with negative resection margins and lymphadenectomy. The regimen of chemotherapy was determined by following the guidelines of the National Health Insurance Service of Korea. The expected accrual period was three years, and the follow-up period was two years in the initial study protocol. This study was performed following the Declaration of Helsinki and Good Clinical Practice guidelines, with each participating institution obtaining approval from their own institutional review board (No.3-2014-0127). The detailed study protocol was published and registered (ClinicalTrials.gov Identifier: NCT01978249) [28].

### 4.2. Participants

Patients older than 20 years who were diagnosed with resectable colon or upper rectal cancer with unresectable metastatic lesions without primary tumor-related symptoms, including intestinal obstruction, intractable bleeding, and perforation were evaluated. Among them, patients with an Eastern Cooperative Oncology Group (ECOG) performance status of 0–2 and an ASA score of less than three were enrolled after providing informed consent. Patients who had mid or low rectal cancer with an anal verge measuring less than 10 cm, unresectable primary cancer, or peritoneal carcinomatosis were excluded from the study. Patients who underwent an incomplete PTR (R2 resection) were also excluded. Unresectable metastases were defined as metastases to two adjacent segments of the liver that could not be preserved after liver resection, a remnant liver volume of <30% of the total liver volume, more than five metastatic lung lesions, requiring a resection larger than a lobectomy, lung metastases resulting in insufficient respiratory function, even if the number of lesions was less than five, or metastases to the brain, bone, neck, mediastinum, or retroperitoneal area. [28].

### 4.3. Sample Size Calculation and Randomization

This study hypothesized that the two-year overall survival rate of Arm 2 would improve by 10% compared to Arm 1 (31% vs. 21%). According to the inequality design, a two-sided log-rank test was performed with an α-error of 0.05 and a power of 80%. The drop-out rate was expected to be 10%. PASS version 12 (NCSS statistical software, Kaysville, UT, USA) was used for the sample size calculation of 240 patients for each group. The patients were allocated into each group according to a 1:1 ratio of randomization, which was stratified by age (<70 vs. ≥70), ECOG performance status (0 vs. 1–2), ASA score (1–2 vs. 3), and target agent (use or no use).

### 4.4. Study Endpoints and Outcome Parameters

The primary endpoint of this study was the comparison of the two-year overall survival rates between the PTR group and the upfront chemotherapy group. The patients were followed up every three months after the initiation of chemotherapy.

The secondary endpoints measured primary tumor-related complication rates, PTR-related complication rates, and rate of conversion to resectable status after chemotherapy. All adverse events after PTR were graded using the Clavien–Dindo classification for surgical complications [29]. Chemotherapy-related toxicities were evaluated based on the National Cancer Institute Common Terminology Criteria for Adverse Events, version 4.0 [30]. PTR-related complications were evaluated in patients of Arm 2, and tumor-related complications were estimated in patients of Arm 1. Neutropenia during chemotherapy was not reported as chemotherapy-related toxicity because it was observed in most patients who received chemotherapy. The rate of conversion of initially unresectable metastatic tumors to surgically resectable tumors during the study period was estimated in both groups.

### 4.5. Data Collection and Monitoring

All clinical data were recorded and collected in the eVelos System, which is a web-based clinical research management system provided by the National Cancer Center, Korea. The final data were confirmed by a principal investigator. The entire process, including data collection, records, and reporting of adverse events, was monitored by a committee that was certificated at each center. The interim results were reported annually to the Ministry of Health and Welfare of Korea.

### 4.6. Statistical Analysis

Statistical analyses were performed using the SPSS 23 (SPSS INC., Chicago, IL, USA) and SAS 9.3 (SAS Institute Inc., Cary, NC, USA) software programs. Categorical variables were compared using the chi-square test or Fisher’s exact test. Continuous variables were analyzed using the Student’s *t*-test or Mann–Whitney U test. The two-year overall survival was estimated using Kaplan-Meier methods. *p* values less than 0.05 were considered as statistically significant.

## 5. Conclusions

PTR followed by chemotherapy improved the two-year cancer-specific survival of patients with asymptomatic stage IV CRC compared with chemotherapy alone. PTR-related major complications were few compared to primary tumor-related complications that developed during chemotherapy. The marginal significance of a *p* value of 0.058 may indicate a possible survival benefit of PTR, although our data did not show an obvious statistical difference in the two-year overall survival. The comprehensive analyses from the results of ongoing randomized controlled trials may provide an answer for the role of PTR followed by chemotherapy in patients with asymptomatic synchronous incurable metastatic CRC.

## Figures and Tables

**Figure 1 cancers-12-02306-f001:**
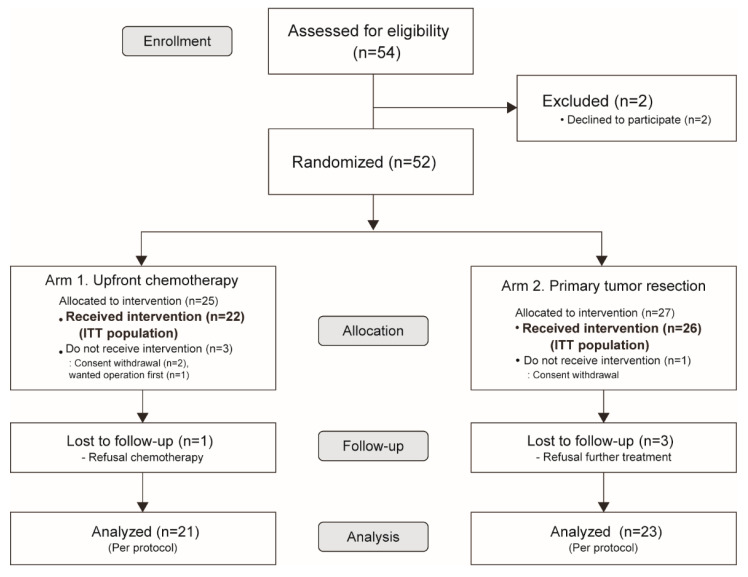
CONSORT flow diagram. ITT, Intention-to-treat.

**Figure 2 cancers-12-02306-f002:**
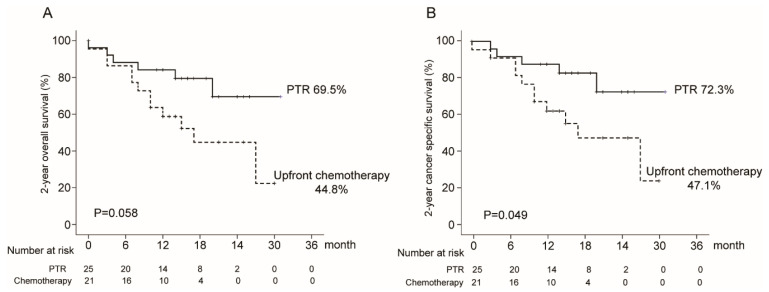
Two-year overall survival: (**A**) Overall survival rate; (**B**) Cancer-specific survival rate.

**Table 1 cancers-12-02306-t001:** Baseline patient characteristics.

	Arm 1 Upfront Chemotherapy Group (*n* = 22)	Arm 2 PTR Group (*n* = 26)	*p* Value
Age, year:	58.8 ± 12.1 (32–74)	62.3 ± 11.8 (40–81)	0.318 ^†^
Sex:			
Male	12 (54.5)	21 (80.8)	0.051 ^‡^
Female	10 (45.5)	5 (19.2)	-
BMI, kg/m^2^:	23.1 ± 2.7 (18.4–28.3)	22.9 ± 3.9 (13.6–34.4)	0.846 ^†^
ASA score:			1.000 ^‡‡^
1	9 (40.9)	12 (46.2)	-
2	10 (45.5)	11 (42.3)	-
3	1 (4.5)	1 (3.8)	-
unknown	2 (9.1)	2 (7.7)	-
ECOG performance status:			0.714 ^‡‡^
0	13 (59.1)	18 (69.2)	-
1	7 (31.8)	7 (26.9)	-
2	2 (9.1)	1 (3.8)	-
Co-morbidity, *n* (%):			0.710 ^‡‡^
Cardiovascular disease	7 (31.8)	9 (34.6)	-
Pulmonary disease	0 (0.0)	2 (7.7)	-
Metabolic disease	2 (9.1)	3 (11.5)	-
None	13 (59.1)	12 (46.2)	-
Location of the primary cancer:			0.706 ^‡‡^
Ascending colon	6 (27.3)	4 (15.4)	-
Transverse colon	1 (4.5)	2 (7.7)	-
Descending colon	0 (0.0)	1 (3.8)	-
Sigmoid colon	11 (50.0)	10 (38.5)	-
Rectosigmoid junction	1 (4.5)	2 (7.7)	-
Rectum	3 (13.6)	7 (26.9)	-
Synchronous metastasis:			0.320 ^‡‡^
Liver only	11 (50.0)	16 (61.5)	-
Lung only	1 (4.5)	2 (7.7)	-
Paraaortic lymph node only	0 (0.0)	2 (7.7)	-
Multiple metastases	10 (45.5)	6 (23.1)	-
Numbers of synchronous metastases:			0.090 ^‡‡^
1	12 (54.5)	20 (76.9)	-
2	10 (45.5)	5 (19.2)	-
≥3	0 (0.0)	1 (3.8)	-
Initial CEA:	121.8 (16.6–1316.1) *	47.6 (26.6–510.6) *	0.649 ^††^

PTR, primary tumor resection; BMI, body mass index; ASA, American Society of Anesthesiologists; ECOG, Eastern Cooperative Oncology Group; CEA, carcinoembryonic antigen. Continuous variables are described as mean ± standard deviation (range); *, median (interquartile range, IQR); categorical variables are described as *n* (%); ^†^ student *t*-test; ^††^ Mann–Whitney U test; ^‡^ Chi-Square test; ^‡‡^ Fisher’s exact test.

**Table 2 cancers-12-02306-t002:** Clinical manifestations in the upfront chemotherapy group.

	Upfront Chemotherapy Group (Arm 1) *n* = 22
Resectable conversion, *n* (%):	4 (18.2)
Liver:	4 (18.2)
Details for the operations of Arm 1:	8 (36.4)
Resectable conversion *:	4 (18.2)
LAR with liver resection	2 (9.1)
AR with liver resection	1 (4.5)
Rt. hemicolectomy with liver resection	1 (4.5)
Palliative PTR **:	2 (9.1)
LAR	1 (4.5)
AR	1 (4.5)
Operations due to intestinal obstruction:	2 (9.1)
Rt. hemicolectomy	1 (4.5)
Laparoscopic AR	1 (4.5)
Intervention:	5 (22.7)
GI stent insertion	3 (13.6)
Salvage radiotherapy	2 (9.1)

LAR, low anterior resection; AR, anterior resection; Rt., right; PTR, primary tumor resection; GI, gastrointestinal; * operation due to conversion to resectable status; ** operation for palliative primary tumor resection.

**Table 3 cancers-12-02306-t003:** Clinical and pathologic manifestations in the PTR group.

	PTR Group (Arm 2) *n* = 26
Operative outcomes for PTR	
PTR procedures:	
Right hemicolectomy	5 (19.2)
Left hemicolectomy	1 (3.8)
Anterior resection	8 (30.8)
Low anterior resection	8 (30.8)
Intersphincteric resection with CAA	2 (7.7)
Subtotal colectomy	1 (3.8)
Total colectomy	1 (3.8)
Operative method:	
Open	16 (61.5)
Laparoscopy	10 (38.5)
Stoma formation:	2 (7.7)
Synchronous operations:	
Segmental resection of small bowel	1 (3.8)
Salpingo-oophorectomy	1 (3.8)
Liver resection, not curative	1 (3.8)
Postoperative clinical outcomes	
Hospital stay, day:	11.8 ± 6.7 (6–34)
Re-admission *:	3 (11.5)
Septic shock	1 (3.8)
General weakness	1 (3.8)
Postoperative ileus	1 (3.8)
Mortality *	1 (3.8)
Time to beginning adjuvant chemotherapy, day	24.8 ± 11.4 (13–66)
Pathologic outcomes of PTR	
T stages:	
T1–2	0 (0.0)
T3	17 (65.4)
T4a/T4b	8 (30.8)/1 (3.8)
Histologic grade of the primary tumor:	
Well-differentiated	1 (3.8)
Moderately differentiated	23 (88.5)
Poorly differentiated	2 (7.7)
Tumor size, cm:	6.0 ± 1.8 (3.4–10.0)
Number of harvested lymph nodes:	26.2 ± 11.1 (8–52)
Number of positive lymph nodes:	5.0 ± 5.0 (0–18)
Lymphovascular invasion:	19 (73.1)
Resectable conversion:	4 (15.3)
Liver	3 (11.5)
Lung	1 (3.8)

PTR, primary tumor resection; CAA, coloanal anastomosis; *, evaluated within 30 days post-operation.

**Table 4 cancers-12-02306-t004:** Comparison of complications between the PTR and upfront chemotherapy groups.

Grade	Upfront Chemotherapy Group (Arm 1) *n* = 22		PTR Group (Arm 2) *n* = 26	
	Primary Tumor-Related Complications	Chemotherapy-Related Toxicity	Postoperative Complications (PTR-Related)	Chemotherapy-Related Toxicity
Grade I	-	6 (27.3%) Pain (1) Skin eruption (3) Nausea (1) HFS (1)	2 (7.7%) Fever (1) Wound seroma (1)	7 (26.9%) Nausea/vomiting (3) Diarrhea (1) Neuropathy (1) HFS (2)
Grade II	-	-	2 (7.7%) Postoperative ileus	-
Grade IIIa	3 (13.6%) Intestinal obstruction	-	-	-
Grade IIIb	2 (9.1%) Intestinal obstruction	-	-	-
Grade IV	-	-	-	1 (3.8%) Pneumonia
Grade V	-	-	1 (3.8%) Pneumonia with sepsis	-

PTR, Primary tumor resection; HFS, Hand-foot syndrome.

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
