# Peer review of "The Role of Primary Tumor Resection in Colorectal Cancer Patients with Asymptomatic, Synchronous, Unresectable Metastasis: A Multicenter Randomized Controlled Trial"

_cancers, 2020, doi:10.3390/cancers12082306_

Round 1

Reviewer 1 Report

The authors Need to be congratulate for the Performance of this RCT.

The results are clear and Support the hypothesis that even in stage IV disease primary tumor resection despite unresectable metatsases is beneficial. The authors could demonstrate this in a small highquality Trial.

Since both groups are relatively small, the effect observed is real.

Some points need to be discussed further:

1) The definition of unresectable mets needs to be pointed out since over 50% of the patinets had only one metatasis (table 1). It seems unlikely that a single met in the liver or lung is unresectable.

2) The authors Need to explain whether they used interventional techniques for local tumor ablation for mets

3) Please explain the type of treatmennt algorithm for rectal cancer patients. Is radiotherapy included? how often was radiotherapy performed, whih protocol was used, etc.

Author Response

We appreciate your invaluable comments, and we have revised our manuscript to address your comments to the best of our ability. Our point-by-point responses to the comments appear below, and changes are shown in red in our manuscript.

The authors need to be congratulate for the Performance of this RCT. The results are clear and support the hypothesis that even in stage IV disease primary tumor resection despite unresectable metastases is beneficial. The authors could demonstrate this in a small high quality Trial. Since both groups are relatively small, the effect observed is real. Some points need to be discussed further:

1) The definition of unresectable mets needs to be pointed out since over 50% of the patients had only one metastasis (table 1). It seems unlikely that a single met in the liver or lung is unresectable.

Response: Thank you for your valuable comment. We agree with your comment that the variables in Table 1 can cause confusion between the numbers of synchronous metastatic organs and lesions. Patients had multiple metastases or a single unresectable metastasis, as follows.

  • After liver resection, two adjacent segments of the liver could not be preserved; vascular inflow, outflow, biliary drainage, or the pedicle could not be preserved; the volume of the remnant parenchyma of the liver was expected to be less than 30% of the total liver volume; or all the metastatic lesions could not be resected completely.
  • There were more than five metastatic lesions in the lung parenchyma; a resection larger than a lobectomy was required, even if there were fewer than five metastatic lesions; or the patient had insufficient respiratory function (i.e., forced expiratory volume < 1 L/s or below 60% of the normal level).
  • Metastases to the brain, bone, neck, mediastinum, or retroperitoneal area.

These specific definitions for unresectable metastases were published in a previous trial (2016:19;17:34. doi: 10.1186/s13063-016-1164-0). Therefore, we have corrected Table 1 to clarify the baseline information for synchronous metastatic organs. In addition, we have added a more detailed definition for unresectable metastases on lines 304–308.

2) The authors Need to explain whether they used interventional techniques for local tumor ablation for mets.

Response: Two patients received salvage radiotherapy for liver metastases, as shown in Table 2. In addition, no patients in this study underwent radiofrequency ablation for liver metastases.

3) Please explain the type of treatment algorithm for rectal cancer patients. Is radiotherapy included? how often was radiotherapy performed, which protocol was used, etc.

Response: In our study, patients with mid and low rectal cancer, which may require pre-/post-operative radiotherapy during treatment, were excluded. The treatment strategies of radiotherapy for locally advanced mid and low rectal cancers are varied and complicated. When we developed the study design, we considered that it would be difficult to randomize patients who need pre-/post-operative radiotherapy between the PTR and upfront chemotherapy groups. Thus, we decided to enroll patients with upper rectal cancer with an anal verge of >10 cm who did not require radiotherapy.

Thank you very much for your beneficial and insightful comments and suggestions. We appreciate that you spent your valuable time in reviewing our manuscript. Although this study was terminated early, we did our best to elucidate whether primary tumor resection is beneficial in patients with asymptomatic, synchronous, unresectable metastases. We hope that our trial results can be helpful in treating stage IV colorectal cancer patients globally.

Sincerely,

Seung Hyuk Baik MD, PhD, FASCRS

Reviewer 2 Report

This is a well conducted study, but unfortunately the authors were not able to succeed in randomizing the appropriate amount of patients needed for a valid conclusion.

However, even with the small numbers of patients there are some interesting findings that are worth publishing. I would suggest to change the conclusion of the manuscript that there is a trend in survival, because the p-value is 0.06.

Author Response

We appreciate your invaluable comments, and we have revised our manuscript to address your comments to the best of our ability. Our point-by-point responses to the comments appear below, and changes are shown in red in our manuscript.

This is a well conducted study, but unfortunately the authors were not able to succeed in randomizing the appropriate amount of patients needed for a valid conclusion.

However, even with the small numbers of patients there are some interesting findings that are worth publishing. I would suggest to change the conclusion of the manuscript that there is a trend in survival, because the p-value is 0.06.

Response: We appreciate your encouragement regarding the study results. As per your comment, the 2-year overall survival rate in the PTR group was marginally significant compared with that in the upfront chemotherapy group, with a p-value below 0.06. However, we had already defined statistical significance as a p-value below 0.05. Therefore, we changed our conclusion (lines 351–352) to include your recommendation as follows: The marginal significance of a p-value of 0.058 may indicate a possible survival benefit of PTR, although our data did not show an obvious statistical difference in the 2-year overall survival.

Thank you very much for your beneficial and insightful comments and suggestions. We appreciate that you spent your valuable time in reviewing our manuscript. Although this study was terminated early, we did our best to elucidate whether primary tumor resection is beneficial in patients with asymptomatic, synchronous, unresectable metastases. We hope that our trial results can be helpful in treating stage IV colorectal cancer patients globally.

Sincerely,

Seung Hyuk Baik MD, PhD, FASCRS

Reviewer 3 Report

This manuscript describes a randomized controlled multicenter study aimed to answer the question whether surgical reaction of the primary tumor before chemotherapy in cases with primary unresectable metastases from colorectal cancer is beneficial.

The scientific question is very relevant and the clinical value of a clear answer to this question is considered to substantially influence the care of these patients. The study protocol is of high quality and I have no major concerns about any step described. Furthermore, the manuscript is generally well written and logically presented.

Unfortunately, the authors failed to include more than one tenth of the intended number of patients in spite of the fact that not less than 12 tertiary south Korean centers were enrolled. The authors mention this as a weakness in their discussion section, but overall, they formulate the discussion from the perspective of valid and reliable results as if the trail had been completed according to the protocol. This is from my perspective not adequate and has to be substantially reformulated.

Nevertheless, given these major concerns about the present design of the manuscript, I am convinced that the authors have much to learn the scientific society. I believe the authors are not the only ones who have designed an almost perfect RCTs targeting this and neighboring groups of patients and failed to reach their targets for several reasons. Without doubt, these complicated patients need and deserve treatment based on high quality research. Why did the study fail to include patients? Which specific advices can the authors give surgeons and oncologists investigating this complicated topic? These questions must be carefully outlined and discussed.

Another interesting and disappointing statement is that the funding ceased. I believe they originally got their funding based on the quality of the protocol. There are many aspects that would benefit from an honest discussion, not least the ethical aspect of having included patients with the view of giving a strong contribution to saving future patients lives, while the attempt to substantially contribute to this failed. From this perspective it is also relevant to comment on the ceased funding and possible alternatives to stop inclusion (for example inviting other international centers?).

A specific example, among many, of statements the have to be revised is the authors explanation why they did not reach significance for differences in overall survival. When reading the manuscript and data, the obvious reason is failing power while all other reasons in this context seem more speculative.

Author Response

We appreciate your invaluable comments, and we have revised our manuscript to address your comments to the best of our ability. Our point-by-point responses to the comments appear below, and changes are shown in red in our manuscript.

This manuscript describes a randomized controlled multicenter study aimed to answer the question whether surgical reaction of the primary tumor before chemotherapy in cases with primary unresectable metastases from colorectal cancer is beneficial. The scientific question is very relevant and the clinical value of a clear answer to this question is considered to substantially influence the care of these patients. The study protocol is of high quality and I have no major concerns about any step described. Furthermore, the manuscript is generally well written and logically presented.

Unfortunately, the authors failed to include more than one tenth of the intended number of patients in spite of the fact that not less than 12 tertiary south Korean centers were enrolled. The authors mention this as a weakness in their discussion section, but overall, they formulate the discussion from the perspective of valid and reliable results as if the trail had been completed according to the protocol. This is from my perspective not adequate and has to be substantially reformulated.

Nevertheless, given these major concerns about the present design of the manuscript, I am convinced that the authors have much to learn the scientific society. I believe the authors are not the only ones who have designed an almost perfect RCTs targeting this and neighboring groups of patients and failed to reach their targets for several reasons. Without doubt, these complicated patients need and deserve treatment based on high quality research.

Response: We appreciate your comments and support regarding this trial.

Why did the study fail to include patients? Which specific advices can the authors give surgeons and oncologists investigating this complicated topic? These questions must be carefully outlined and discussed.

Response: Thank you for your comment. We think that these questions are invaluable for clinical investigators conducting a trial in stage IV cancer patients. We encountered many difficulties in conducting this trial, from the points of view of patients, clinicians, and participating institutions.

During this study, it was difficult to reach an agreement in terms of patient enrollment. Most patients were depressed after the diagnosis of synchronous unresectable metastases from colorectal cancer. Most patients hesitated to participate in this clinical trial when they were informed that they would be randomized to undergo surgery or receive chemotherapy. Some patients were afraid to undergo surgery immediately because their primary colorectal cancer was asymptomatic. In addition, although the detailed inclusion criteria were beneficial in ensuring the high quality of the clinical trial, the use of these criteria made patient enrollment difficult.

Furthermore, participating surgeons also faced difficulties performing PTR within 14 days of patient enrollment and randomization. In South Korea, most surgeons in tertiary hospitals perform many colorectal cancer surgeries as well as emergency operations. Although most surgeons agreed on the value of this study, they were already tired of conducting their routine jobs. Therefore, participation may have decreased owing to the difficulties faced by surgeons in explaining this trial, actively enrolling patients, and performing PTR. In addition, most oncologists preferred not PTR but upfront chemotherapy in patients with asymptomatic stage IV colorectal cancer. It was difficult to justify this trial and our purpose to oncologists who had negative opinions regarding PTR.

Lastly, active participation of each institution in a clinical trial is essential to achieve the final goal and successful enrollment. We attempted to include the major hospitals in our country, which are all members of the Korean Society of Coloproctology (KSCP). 

We made efforts to achieve patient enrollment with regular meetings and communication with the participants of this trial. However, it was difficult to achieve successful enrollment despite these efforts. We consider that surgical intervention for stage IV colorectal cancers was challenging for both patients and surgeons. We have added information on these difficulties in patient enrollment in the discussion (lines 258–263).

For the above reasons, it was difficult to successfully complete this study despite our best efforts. However, we undertook this trial to elucidate whether PTR is beneficial in improving survival and preventing complications. As colorectal surgeons, our daily work is to provide consultation and treatment to cancer patients in the best possible way. In the treatment of patients with unresectable metastases of asymptomatic colorectal cancer, the role of PTR is questionable because there is no level I evidence thus far. Therefore, it is important to elucidate whether PTR is beneficial for stage IV colorectal cancer patients. We believed that it is not only our wishes but also our mission to identify the best treatment strategy for surgery in stage IV patients.

Another interesting and disappointing statement is that the funding ceased. I believe they originally got their funding based on the quality of the protocol. There are many aspects that would benefit from an honest discussion, not least the ethical aspect of having included patients with the view of giving a strong contribution to saving future patients’ lives, while the attempt to substantially contribute to this failed. From this perspective it is also relevant to comment on the ceased funding and possible alternatives to stop inclusion (for example inviting other international centers?).

Response: Thank you for your important comment regarding research funding. This trial was supported by the Korean National Cancer Control Planning Board Study, which is funded by the Ministry of Health and Welfare and is managed by the National Cancer Center of South Korea. Initially, this study was designed to last 5 years, from 2013 to 2018. The research fund was paid through the regular annual assessment. However, in 2016, funding ceased abruptly due to the lack of enrollment. Although we made efforts to continue this trial, it was impossible because of the difficulties in performing randomization and data collection using the eVelos system, which should be supported by research funding. We believe that research funding is required to support a clinical trial during the study period and the waiting period for sufficient study enrollment, which is achieved later. Study enrollment is slower in the initial stage than in the later stages. The abrupt cessation of funding was a major obstacle to completing this study. As per your comment, we agree that both collaboration with international centers and finding other sources of research funding can be alternatives means of completing this trial. We will consider your recommendations in our future clinical trials. We have added these opinions to the discussion section (lines 263–264).

A specific example, among many, of statements have to be revised is the authors explanation why they did not reach significance for differences in overall survival. When reading the manuscript and data, the obvious reason is failing power while all other reasons in this context seem more speculative.

Response: Thank you for your comment. We agree that this study is limited by low statistical power due to the lack of enrollment. However, we did our best to collect data and conclude our study with the available enrolled patients. According to this trial, PTR plus chemotherapy did not demonstrate an improved 2-year overall survival rate compared with chemotherapy alone. However, we think that the marginal significance represented by a p-value of 0.058 may indicate a possible survival benefit of PTR. In future studies, global collaboration is expected to help elucidate the survival benefit of PTR. We have added this information on lines 351–352.

Thank you very much for your beneficial and insightful comments and suggestions. We appreciate that you spent your valuable time in reviewing our manuscript. Although this study was terminated early, we did our best to elucidate whether primary tumor resection is beneficial in patients with asymptomatic, synchronous, unresectable metastases. We hope that our trial results can be helpful in treating stage IV colorectal cancer patients globally.

Sincerely,

Seung Hyuk Baik MD, PhD, FASCRS

Reviewer 4 Report

I have the following comments and suggestions that the authors may wish to consider in the revision:

Major comments:

  • For patients with unresectable distant metastases who have symptoms from the primary tumor, there is no question about considering primary tumor resection (PTR) as the highest priority. However, there is no consensus on treatment for asymptomatic cases that account for more than half of incurable stage IV colorectal cancer. As the authors mentioned in the discussion, the early termination of this study resulted in a lack of date to help the physicians make clinical decision whether PTR should be performed or not for the patients with asymptomatic primary tumor and unresectable metastases.
  • Why was the primary endpoint 2-year overall survival? It is unusual.
  • It seems that a regimen of chemotherapy depended on the physician’s choice. However, it should have been decided by investigators before randomization, considering the generalization of the results.

Author Response

We appreciate your invaluable comments, and we have revised our manuscript to address your comments to the best of our ability. Our point-by-point responses to the comments appear below, and changes are shown in red in our manuscript.

I have the following comments and suggestions that the authors may wish to consider in the revision: For patients with unresectable distant metastases who have symptoms from the primary tumor, there is no question about considering primary tumor resection (PTR) as the highest priority. However, there is no consensus on treatment for asymptomatic cases that account for more than half of incurable stage IV colorectal cancer. As the authors mentioned in the discussion, the early termination of this study resulted in a lack of date to help the physicians make clinical decision whether PTR should be performed or not for the patients with asymptomatic primary tumor and unresectable metastases.

Why was the primary endpoint 2-year overall survival? It is unusual.

Response: In this study, we evaluated stage IV colorectal cancer patients with synchronous unresectable metastases. As the mean survival rates in stage IV metastatic colorectal cancer patients were 20–35%, we considered the 2-year overall survival endpoint acceptable for evaluating the differences in survival between PTR and upfront chemotherapy. We also referred to previously published studies that evaluated the 2-year overall survival to compare PTR and upfront chemotherapy.

It seems that a regimen of chemotherapy depended on the physician’s choice. However, it should have been decided by investigators before randomization, considering the generalization of the results.

Response: According to our initial study design, we recommended that each center choose a chemotherapy regimen as shown in the figure (Please see the attachment). Although most centers followed our protocol, some patients were treated by capcitabine or XELOX because of the patients’ conditions or adverse effects. We accepted each doctor’s decision for the choice of chemotherapy regimen because the investigators at each institution have been specialists in colorectal cancer treatment for over 20 years. As per your comment, the variation in chemotherapy regimens can be a limitation of our study. Therefore, we have included this information in the discussion (lines 268–270).

Thank you very much for your beneficial and insightful comments and suggestions. We appreciate that you spent your valuable time in reviewing our manuscript. Although this study was terminated early, we did our best to elucidate whether primary tumor resection is beneficial in patients with asymptomatic, synchronous, unresectable metastases. We hope that our trial results can be helpful in treating stage IV colorectal cancer patients globally.

Sincerely,

Seung Hyuk Baik MD, PhD, FASCRS

Round 2

Reviewer 3 Report

The authors have adequately and sufficiently responded to the questions raise and improved the manuscript accordingly.

Reviewer 4 Report

Although the small sample size due to early termination, PTR tended to be a better treatment in terms of OS than chemotherapy alone. Further studies are warranted for colorectal cancer patients with an asymptomatic primary tumor and synchronous unresectable metastases.